# Mean Field Approaches to Lattice Gauge Theories: A Review

**DOI:** 10.3390/e27030250

**Published:** 2025-02-27

**Authors:** Pierpaolo Fontana, Andrea Trombettoni

**Affiliations:** 1Departament de Física, Universitat Autònoma de Barcelona, 08193 Bellaterra, Spain; 2Department of Physics, University of Trieste, Strada Costiera 11, I-34151 Trieste, Italy; 3SISSA and INFN, Sezione di Trieste, Via Bonomea 265, I-34136 Trieste, Italy

**Keywords:** lattice gauge theory, mean field theory, gauge-invariant approaches

## Abstract

Due to their broad applicability, gauge theories (GTs) play a crucial role in various areas of physics, from high-energy physics to condensed matter. Their formulations on lattices, lattice gauge theories (LGTs), can be studied, among many other methods, with tools coming from statistical mechanics lattice models, such as mean field methods, which are often used to provide approximate results. Applying these methods to LGTs requires particular attention due to the intrinsic local nature of gauge symmetry, how it is reflected in the variables used to formulate the theory, and the breaking of gauge invariance when approximations are introduced. This issue has been addressed over the decades in the literature, yielding different conclusions depending on the formulation of the theory under consideration. In this article, we focus on the mean field theoretical approach to the analysis of GTs and LGTs, connecting both older and more recent results that, to the best of our knowledge, have not been compared in a pedagogical manner. After a brief overview of mean field theory in statistical mechanics and many-body systems, we examine its application to pure LGTs with a generic compact gauge group. Finally, we review the existing literature on the subject, discussing the results obtained so far and their dependence on the formulation of the theory.

## 1. Introduction

Gauge theories (GTs) have a key importance in physics that can hardly be overstated [1,2,3]. In the field of particle physics, they form the basis of the Standard Model, a non-Abelian gauge theory with gauge group U(1)×SU(2)×SU(3), where the first two groups refer to the electroweak sector and the last one governs quantum chromodynamics (QCD). They also play an important role in condensed matter physics, where gauge fields often arise in effective descriptions of strongly correlated phenomena at low energies, such as quantum Hall systems and quantum spin liquids [4].

The discretization of GTs on a lattice is one possible way to deal with the strongly coupled nature of these theories [5,6], because, in a finite volume, it provides natural regularizing infrared and ultraviolet cut-offs. It also allows for the investigation of different non-perturbative phenomena, both numerically and analytically [7,8]. In this respect, Monte Carlo methods stand out as a very powerful tool to extract information from lattice gauge theories (LGTs), and crucial results have been obtained in the literature, in particular in lattice QCD. Despite the remarkable amount of progress, various aspects remain intractable due to sign problems or complex action problems, such as out-of-equilibrium real-time evolution or the analysis of QCD with finite chemical potential [9,10].

Concerning analytical methods, statistical mechanics techniques, such as series expansions and mean field (MF) approximations, are powerful tools for obtaining approximate results [11]. When applying these methods to LGTs, particular attention must be paid to the intrinsic local nature of gauge symmetry and how it is reflected in the formulation of the theory, given that approximate methods typically break gauge invariance. This challenge has been explored in the literature, leading to different conclusions depending on the specific formulation of the underlying theory. An important point to note is that different formulations of a GT, or an LGT, obtained by mapping the original theory to a different set of variables will give the same results for the physical observables when the theory is exactly treated, but can—and actually will—give different results when subjected to an approximate method, such as the MF. Moreover, one formulation may perform significantly better than another under the same approximation scheme.

These issues can also be present for static gauge fields. As an example, we can consider the analysis of the electromagnetic response of superconductors in background electromagnetic fields, treated in the framework of BCS theory [12]. In a fully exact treatment of the electromagnetic response, gauge invariance follows directly from local charge conservation. However, by introducing the MF approximation, the charge conservation may be broken, therefore leading to results that depend on the specific gauge. This suggests that the simple pairing scheme of the BCS theory may not account for these effects in the presence, e.g., of a magnetic field. More sophisticated, gauge-invariant approaches have been proposed over the years to describe the electromagnetic properties of superconductors while keeping gauge invariance in approximate methods, such as the MF one [12,13,14,15].

In this paper, we focus on LGTs, although several of our results and discussions may apply to GTs as well. We present various formulations of LGTs in terms of gauge-invariant variables with the goal of applying MF methods while retaining gauge invariance within the chosen approximation. As we will explain in more detail later, these formulations are designed to make the variables in the theory consistent with Elitzur’s theorem [16], which dictates which expectation values are compatible with the gauge symmetry. In particular, using the original approach to discretize GTs on a lattice [5], the resulting link variables are not immediately suitable for MF applications, as their expectation value is zero. Several solutions to this issue have been proposed in the literature, and our purpose is to provide a pedagogical overview of this topic and its proposed solutions, connecting traditional approaches with more modern variational techniques, a discussion which we believe may interest researchers working on LGTs, interacting lattice models, and quantum simulations of different types of LGTs [17,18,19,20].

The structure of this paper is as follows: In Section 2, to set the stage for the subsequent chapters, we review the MF theory in its variational form, showing simple and well-known examples of its application to lattice spin models and to condensed matter field theories, as a concrete instance. Section 3, Section 4 and Section 5 give the bulk of the information presented in this paper. In Section 3, following the discussion presented in [21], we first recall what a gauge theory is, introducing the relevant quantities in the continuum. In Section 4 we present a discussion on LGTs, both in the Lagrangian and Hamiltonian formulations, introducing bosonic and fermionic link models. We then state Elitzur’s theorem, which is crucial for the application of MF methods. In Section 5, we introduce various reformulations of LGTs which have been introduced over the years, discussing their similarities and differences. In Section 6, we outline how to apply the MF approximation to LGTs, highlighting the advantages and disadvantages of different reformulations and reviewing results for pure LGTs with compact gauge groups. Finally, in Section 7, we present our conclusions.

## 2. Reminder on Mean Field Theory

In this section, we give a brief reminder of the MF method, with the aim of applying it to statistical lattice models. There are different ways to derive the MF approximation, and here, we present its variational formulation, which is suitable for application to LGTs [8,11].

Let us consider a generic lattice field theory, formulated in terms of a field ϕ, with values defined in a vector space *V*. We denote the action of the system with S[ϕ], and the associated partition function, *Z*, can be formally written as(1)Z(β)=e−βF=∫Dϕe−S[ϕ],Dϕ≡∏x∈Λdϕx
where β=1/kBT is the inverse temperature, x∈Λ specifies the lattice coordinate, and *F* is the free energy of the system. The general idea behind the MF approximation is to replace the dynamics of the full theory by that of independent degrees of freedom in a given external source, called the *mean field*, which must be chosen in a way to simulate the real dynamics in the best possible way.

To realize this, among many other ways [22,23,24], one may add and subtract a *source term Sh[ϕ]* in the action(2)S[ϕ]→S[ϕ]+Sh[ϕ]−Sh[ϕ]≡S˜h[ϕ]+Sh[ϕ],
where *h* plays the role of a variational parameter. The partition function assumes the form(3)Z=∫Dϕe−S˜h[ϕ]−Sh[ϕ]≡Zh〈e−S˜h[ϕ]〉h,
where we denote all the modified quantities throughout the introduction of the source term, defined as(4)Zh≡∫Dϕe−Sh[ϕ],〈O〉h≡1Zh∫DϕOe−Sh[ϕ],
with the subscript *h*. To take advantage of this alternative form of the partition function, we apply the Jensen inequality to the exponential function [25]. This inequality states that, given a real function f(x) and a normalized measure ρ(x),(5)〈ef〉≥e〈f〉,〈f〉≡∫f(x)ρ(x)dx.
We can use this inequality, which in MF theory is also known as the *Bogoliubov inequality*, to bound the exact free energy of the model from below. In our case, the measure ρ(ϕ) can be read immediately from the expectation value in Equation (4).

Therefore, the application of Equation (5) to the partition function *Z* gives(6)ZZh=〈e−S˜h[ϕ]〉h≥e−〈S˜h[ϕ]〉h,
and passing to the free energy densities through the logarithm, since F≡−β−1logZ, we obtain(7)βF≤βFh+〈S˜h[ϕ]〉h,
The optimal estimate in terms of the variational parameter *h*, i.e., the value of *h* that saturates the bound in Equation (7), leads to the relation(8)βFMF=minh[βFh+〈S˜h[ϕ]〉h],
which gives a self-consistent equation for the external field *h*. Expressed this way, the application of the self-consistent Equation (8) is convenient for the Lagrangian formulation of LGTs, as will be outlined in the following sections.

We finally point out that we did not specify the form of the source term leading to the modified action S˜h[ϕ]. This is because there is no general method to choose it, a point that represents a major challenge in the MF procedure. The only reasonable requirement is that they should lead to a computable form of the Equation (8), allowing us to extract useful information from the modified theory. This is why a non-interacting source term is usually considered.

### 2.1. An Application to Spin Models

We apply the MF argument to a simple spin model, to show a concrete application of the variational method just introduced [23,25,26]. As a prototypical system, we consider the ferromagnetic Ising model in *d* dimensions, on a hypercubic lattice Λ, with Hamiltonian(9)HIsing=−J∑〈i,j〉sisj,si±1,
and in the following, we consider J>0. This Hamiltonian describes the interactions between two nearest-neighbor spins located at sites i,j∈Λ. We point out that the field ϕ is now replaced by a discrete spin variable with values in Z2, which means that all the integrals in Dϕ are replaced with discrete sums over all possible spin configurations {s}.

We can now write down the modified Hamiltonian by adding and subtracting a non-interacting source term of the form(10)HIsing→H˜Ising−h∑i=1Nsi,H˜Ising=−J∑〈i,j〉sisj+h∑i=1Nsi,
where *N* is the number of lattice sites and *h* is the mean field. Physically, the source term is the Hamiltonian of a simple paramagnet, whose free energy density and expectation value of the spin variable are known and equal to(11)Fh=−1βlog(2cosh(βh)),〈s〉h=tanh(βh).
The last quantity, 〈s〉h, turns out to be of fundamental importance, since it allows us to distinguish between the different phases of the theory, depending on the solutions of Equation (8). For the Ising model, this is called the magnetization *m*. In the Landau theory of phase transition, a quantity with this property is called the *order parameter* [1,2,3,27,28]. According to Equation (8), the other quantity we need to know is(12)〈H˜Ising〉h=−J∑〈i,j〉〈si〉h〈sj〉h+h∑i=1N〈si〉h=−JdN〈s〉h2+Nh〈s〉h,
where in the last step, we assumed translational invariance, reflecting into 〈si〉h=〈sj〉h≡〈s〉h. Inserting these quantities into Equation (8), we end up with the free energy density(13)βF=βhtanh(βh)−log[2cosh(βh)]−Jdβtanh2(βh)
This quantity contains different terms, depending on the value of the inverse temperature: the first two terms tend to disorder the system, trying to force the external source to vanish; on the other hand, the last term is trying to force h≠0, and can be interpreted thermodynamically as a potential energy.

To better discuss the last statement, we apply the variational form of the MF approximation. From Equation (7), we read that FMF≤F, and the saturation of this bound can be obtained by solving the self-consistent Equation (8), giving the effective values of the external sources at finite β:(14)∂(βF)∂h=h−2Jdtanh(βh)cosh2(βh)=0.
The same equation can be rephrased in terms of the magnetization as(15)h=2Jdtanh(βh)⇔m=tanh(2βJdm),
where we used h=2dJm, coming from the expectation value 〈s〉h in Equation (11). Finally, inserting Equation (15) into the free energy density, we find the MF free energy FMF as(16)βFMF=Jdβm2−log[2cosh(2βJdm)].

The solution to the MF self-consistent equation for different values of the inverse temperature β gives the possible MF values of *m*. In Figure 1 (left plot), we analyze the different possible cases. For small values of β, there are no solutions to Equation (15) apart from the trivial one, i.e., m=0. This is consistent with the fact that at high temperatures, the system is *disordered*, and the average magnetization is zero. For large values of β, the system can develop a non-trivial order parameter, i.e., m≠0. This means that, at low temperatures, the system is *ordered* and has a spontaneous magnetization different from zero. The two regimes are connected through a critical value, denoted by βMF, for which the two gradients of both sides of Equation (15) are equal at the origin. This allows us to precisely identify the critical value as βMF=1/2dJ.

Additionally, in Figure 1 (right plot), we present the behavior of the MF free energy density βFMF as a function of the previously discussed solutions for the order parameter *m*. For β<βMF, the free energy has one minimum at m=0, while there are two minima in the opposite limit of β>βMF. These minima represent the two states of the system with spontaneous magnetization m≠0, which are energetically equivalent due to the Z2 symmetry of the model. As before, the two cases are connected by the curve at β=βMF.

### 2.2. An Application to Many-Body Systems: The BCS Theory

As a second example, we move to the realm of condensed matter and remind of the standard MF description of superconductors, introducing the Bardeen–Cooper–Schrieffer (BCS) theory from the microscopic point of view [12,27,29].

Starting from the BCS Hamiltonian,(17)HBCS=∑k,σϵkckσ†ckσ−gV∑k,k′,qck+q,↑†c−k,↓†c−k′+q,↓ck′↑
where σ=↑,↓ is the spin degree of freedom, g>0 is the coupling constant, ϵk=ℏ2k2/2m is the single particle energy spectrum (measured with respect to the Fermi energy) and ck†,ck are momentum space fermionic creation and annihilation operators; we can employ the path integral formalism to derive the MF equations [27]. We write the associated BCS Lagrangian in terms of the fermionic field in real space ψσ(r) as(18)LBCS=∑σ∫drψσ†(r)∂τ−∇22m−μψσ(r)−g∫drψ↑†(r)ψ↓†(r)ψ↓(r)ψ↑(r)
and the path integral representation of the partition function in Equation (1) written in terms of coherent states [1,3], with action(19)S[ψ¯,ψ]=∫0βdτ∫drψ¯σ∂τ−∇22m−μψσ−gψ↑†ψ↓†ψ↓ψ↑,
where ψ(r,τ) are Grassmann fields. The first step towards the derivation of the MF free energy is to introduce the order parameter, which is usually achieved by means of a Hubbard–Stratonovich transformation. This means writing the interaction term as an integral over an auxiliary field Δ, that is,(20)exp∫0βdτ∫drgψ↑†ψ↓†ψ↓ψ↑=∫D[Δ,Δ¯]exp−∫0βdτ∫dr−|Δ|2g−Δ¯ψ↓ψ↑+Δψ¯↑ψ¯↓.
The dynamical field Δ=Δ(r,τ) is assumed to be bosonic, satisfying the periodic boundary condition Δ(τ+β)=Δ(τ) in imaginary time. The integration measure D[Δ,Δ¯], besides the products of the two measures DΔ and DΔ¯ defined in Equation (1), is conveniently defined to absorb the numerical prefactor coming from the Gaussian functional integration [30].

By introducing the Nambu spinor Ψ¯≡(ψ¯↑,ψ↓), we can recast the partition function in a more compact notation, as the functional integral(21)Z=∫D[Ψ,Ψ¯]∫D[Δ,Δ¯]exp∫0βdτ∫dr−|Δ|2g−Ψ¯G−1Ψ,
where the Green function G has been defined as(22)G−1≡G0−1+Σ=G0,p−100G0,h−1+0ΔΔ*0,G0,[p,h]−1=−∂τ±∇22m±μ.
The expression of G0−1 can be read off from the first term of the action for the non-interacting particle (p) and hole (h) Green functions. Since the action is bilinear in the Nambu spinor, we can formally perform the Gaussian integration over the Grassman variables, generating the determinant of the matrix G−1. The final effective action is(23)Seff[Δ,Δ¯]=∫0βdτ∫dr|Δ|2g−Trlog(G−1).

We are now in the position of performing the MF approximation for this effective action written in terms of the field Δ. To simplify the discussion, we assume that the auxiliary field Δ is constant in space and time, Δ(r,τ)≡Δ, and perform the saddle-point(24)δSeffδΔ*=0⇒Δg−TrGδG−1δΔ*=0,
which is a self-consistent equation in the parameter Δ, a condition analogous to the Equation (8). From Equation (22), we have that the variation of the Green function with respect to Δ gives rise to a matrix in the space of the Nambu spinors which has a single element different from zero, in correspondence with the position of Δ*. Given the inverse of the matrix G−1, the self-consistent equation reduces to(25)G=−1G0−1+|Δ|2−[G0−1]†−Δ−Δ*G0−1⇒1g=1(G0−1)2+|Δ|2.
This is the celebrated BCS gap equation, in the form derived from the path integral formulation of the microscopic theory. It can be conveniently rewritten by switching to momentum and frequency representation, where ∂τ→−iωn, −∇2→p2 and the matrix G has the form(26)G(p,ωn)=−1ωn2+ω˜2iωn+ϵp−μΔΔ*iωn−ϵp−μ,ω˜2≡(ϵp2−μ)2+|Δ|2,
with ϵp representing the single-particle energy of the non-interacting system, as given in Equation (17). The gap equation becomes(27)1g=1β∑p,n1ωn2+ω˜2=∑p1−2nFD(ω˜)2ω˜=∑p12ω˜tanhβω˜2,
where, in order, we performed the sum over the Matsubara frequencies ωn, and introduced the Fermi–Dirac distribution nFD(ω˜), satisfying the property 1−2nFD(ω˜)=tanh(βω˜/2) [27].

This self-consistent equation for the constant field Δ plays the same role as the Equation (15) in the Ising model. The pairing is self-consistently determined by the interactions between fermions. The constant field Δ now represents the gap of the BCS theory. The spectrum of the superconductor is defined by the quantity ω˜=(ϵp2−μ)2+|Δ|2. The hyperbolic tangent factor includes the dependence on the temperature, which is in β=1/kBT. This information is encoded in a single equation, obtained through the saddle-point approximation, showing the well-known fact that, despite its formal simplicity, the MF theory captures the relevant features of the physics of BCS superconductors [12].

## 3. Gauge Symmetry in Quantum Field Theory

To define what GTs are, we consider as a starting point a multi-component scalar field ϕ=(ϕ1,…,ϕn)T with values in the vector space V=Cn, whose Lagrangian and Lorentz-invariant action are(28)L[ϕ,∂μϕ]=∂μϕ*·∂μϕ−m2ϕ*·ϕ,S=∫ddxL[ϕ,∂μϕ].
We observe that the Lagrangian is invariant under global U(n) transformations of the scalar field(29)ϕ(x)⟶ϕ′(x)=Ωϕ(x),Ω∈U(n),
where Ω does not depend on the space–time point coordinates. At the same time, the scalar product is invariant under simultaneous U(n) transformations, since ϕ*·χ=(ϕ′)*·χ′. However, the Lagrangian is not invariant under the corresponding *local* transformation(30)ϕ(x)⟶ϕ′(x)=Ω(x)ϕ(x),Ω(x)∈U(n).
It is possible to extend the above global symmetry to a local one, by coupling the charged scalar field ϕ to a gauge potential Aμ [1,2,3,26]. In the so-called minimal coupling procedure, the space–time derivative is replaced by the covariant derivative(31)∂μ⟶Dμ(A)=∂μ−igAμ,
where *g* is the coupling constant of the matter–gauge interaction. By requiring that the covariant derivative locally transforms as(32)Dμ(A′)ϕ′(x)=Ω(x)Dμ(A)ϕ(x),
one finds that, using Equation (30), this reduces to the condition(33)Dμ(A′)=Ω(x)Dμ(A)Ω(x)−1.
This equation is satisfied if the gauge potential transforms as(34)Aμ′(x)=Ω(x)Aμ(x)Ω(x)−1−ig[∂μΩ(x)]Ω(x)−1.
If we assume that Ω(x)∈G, where G is a Lie group, then all the other elements appearing in Equation (34) are Lie-algebra valued.

Out of the gauge potential, we can construct the field strength tensor as the commutator(35)Fμν(A)=ig[Dμ,Dν]=∂μAν−∂νAμ−ig[Aμ,Aν],Fμν⟶Ω(x)FμνΩ(x)−1.
In analogy to quantum electrodynamics (QED), we can consider the square of Fμν to obtain the dynamical term of the gauge field. However, if the gauge symmetry group is non-Abelian, the quantity FμνFμν is not gauge-invariant. As is well known, if we take its trace, i.e., tr(FμνFμν), we generalize the QED term to the Yang–Mills term.

The full gauge-invariant Lagrangian is then(36)LG[ϕ,Aμ]=−14tr(FμνFμν)+L[ϕ,Dμϕ].
We notice that, apart from the Yang–Mills term for the gauge part, we have the initial Lagrangian in Equation (28) with the minimal substitution ∂μ→Dμ(A).

From now on, we will consider the specific case of Abelian groups, referring to Abelian GTs. Formally, the previous local transformations can be written as(37)ϕ(x)→Ω(x)ϕ(x),Aμ→Aμ−ig[∂μΩ(x)]Ω−1(x)
for the matter and gauge fields, respectively. The field strength tensor Fμν takes the form(38)Fμν≡∂μAν−∂νAμ.
Under a gauge transformation, this quantity is left unchanged, i.e., Fμν→Fμν; therefore, it represents a *gauge invariant* of the theory. A prominent example of an Abelian gauge theory is QED. Its Lagrangian reads(39)LQED=ψ¯(iD−m)ψ−14FμνFμν=ψ¯(i∂−m)ψ−eAμψ¯γμψ−14FμνFμν,
where ψ and ψ¯ are the fermionic degrees of freedom and *m* and *e* are, respectively, the fermionic mass and charge parameters. The gauge group is G=U(1) and a generic local transformation can be written as a phase factor Ω(x)=expieΛ(x).

### 3.1. Equations of Motion and Bianchi Identity

From the Lagrangian in Equation (36), we can write down the equations of motion for the vector potential Aμ, to see how the local symmetry influences its dynamics, both in the absence and presence of the scalar field ϕ [1,2,3]. The Euler–Lagrange equations for LG read(40)∂LG∂Aν−∂μ∂LG∂(∂Aμ)=0,
where(41)∂LG∂Aν=ig(ϕ*Dνϕ−(Dνϕ)*ϕ)≡gjν,
where jν is the Noether current associated with the matter field ϕ, and(42)∂μ∂LG∂(∂Aμ)=∂μFμν.
This results in the inhomogeneous field equations(43)∂μFμν=gjν.
In the case of QED, these represent the covariant form of the Maxwell equation with dynamical sources. Alongside them, we can also derive the equations of motion for the matter field, writing down the variations in the Euler–Lagrange equations with respect to ϕ itself, ending up with the Klein–Gordon equations with minimal coupling, i.e., DμDμϕ=0. Taking the temporal component of Equation (43), fixing ν=0, we have(44)∂μFμ0=∂iFi0=gj0⇒∇·E=gρ,
where ρ=j0 is the charge density, and we identified Fi0 with the electric field Ei. This is the well-known Gauss law for the electric field in its differential form. This equation is different if non-Abelian gauge fields are considered. The divergence of the electric field ∇·E is replaced with ∇·E+[A,E], highlighting that the non-Abelian vector potential contributes to the charge density.

As a final feature, we mention that the definition of Fμν has important consequences from the geometrical point of view. Since the field strength tensor is defined as Fμν≡∂μAν−∂νAμ, the following identity holds, called the *Bianchi identity*:(45)∂τFμν+∂μFντ+∂νFτμ=0,
obtained by taking the partial derivative of Fμν and summing over the cyclic permutations of the indices μ,ν,τ. In the case of QED, this equation encodes two fundamental laws of electromagnetism, namely the Gauss law for the magnetic field (∇·B=0) and Faraday’s law (∇×E+∂tB=0).

More technically, in differential geometry, the field strength tensor Fμν is associated with a two-form *F*, which is the *exterior derivative* of the one-form *A* of the vector potential Aμ, i.e., F=dA. The fact that dF=0, i.e., it is a *closed form*, represents the Bianchi identity. In tensor form, i.e., explicitly for the field strength tensor Fμν, this results in Equation (45). In the Abelian case, this constraint is fundamental for Fμν, due to its antisymmetry in the space–time indices, and technically does not require Aμ (this is fundamentally different in the non-Abelian case, as the Bianchi identity involves the vector potential. Indeed, Equation (45) is replaced with [DτFμν]+[Dμ,Fντ]+[DνFτμ]=0, involving a commutator with the covariant derivative. Since the latter involves Aμ directly, this relation requires the existence of the vector potential). However, there is always the possibility of choosing Aμ that can locally give Fμν. This last statement is a consequence of an important lemma in differential geometry, known as the Poincaré lemma [31].

### 3.2. Parallel Transport

A key object in the passage to the lattice formulation of a gauge theory is the parallel transport. This can be defined in the continuum, and on the lattice, its discretized version is a fundamental constituent of the theory. We start with the definition of a covariantly constant field, i.e., a field ϕ that satisfies(46)Dμϕ=0⇒[Dμ,Dν]ϕ=0.
This equation can be studied along a path Cxy joining two space–time points x,y. We can parametrize this path by z(s), with s∈[0,1] and z(0)=x,z(1)=y. The field ϕ is said to be covariantly constant along the path Cxy if(47)z˙μDμϕ=0⇒ϕ˙(s)−igAμ(z(s))z˙μ(s)ϕ(s)=0,
where ϕ(s)≡ϕ(z(s)). The solution to this equation can be obtained by direct integration:(48)ϕ(s)=Peig∫0sduAμ(z(u))z˙μ(u)ϕ(x),
where P stands for the path-ordering operation. This symbol is necessary in the non-Abelian case, while is superfluous for Abelian theories, due to the commutativity of the symmetry group. The definition of parallel transporter can be obtained by putting s=1 into the previous equation(49)ϕ(y)=UCxy(A)ϕ(x),UCxy(A)=Peig∫01dsAμz˙μ.
The parallel transport is, by definition, a path-dependent quantity, as it is a function of Cxy. It has the following properties:If C1:x→y and C2:y→z are two space–time paths, it satisfies the composition rule(50)UC1○C2=UC1UC2;Under a gauge transformation(51)UCxy(A′)=Ω(y)UCxy(A)Ω(x)−1.
From the last property, it follows immediately that the trace of the parallel transporter associated with an arbitrary closed loop is a gauge-invariant quantity, the Wilson loop, and is defined as(52)WCxy(A)≡tr[UCxy(A)].
As anticipated, this is a gauge-invariant quantity since tr[UCxy(A′)]=tr[UCxy(A)].

## 4. Lattice Gauge Theories

The analysis of GTs in the strong coupling regime is an arduous problem where perturbative approaches typically may fail. One way to deal with this problem is to work in the framework of LGTs [5,6,7,32]. The lattice formulation at finite volume provides natural infrared and ultraviolet cut-offs that regularize the theory. Moreover, within this formulation, numerical approaches to the problem are possible using Monte Carlo methods [7,32] and crucial results have been obtained, e.g., for lattice QCD in its strongly coupled low-energy regime. Among the various points that have been addressed, we mention here the studies regarding string tension and quark potentials in pure GTs and full QCD [33,34,35,36], chiral symmetry breaking [37], the mass spectrum of bound states in pure QCD [38], the hadron mass spectrum [39,40], and the deconfinement phase transition and high-temperature phases of QCD [41,42].

The lattice discretization of GTs can be performed following essentially two paths. The first one entails the discretization of the continuum theory Lagrangian. This constitutes the *Lagrangian formalism* of LGTs. There is also the possibility of considering the *Hamiltonian formalism*, in which space dimensions are discretized but time is not. In this formulation, the theory is projected only on its physical states |Ψ〉, i.e., the ones satisfying Gauss law [6,17]. In these discretization schemes involving fermions, it is well known that particular attention must be paid to address the fermion doubling problem [7]. This can be achieved by considering different discretizations of the fermionic field (e.g., Wilson fermions, staggered fermions, or domain wall fermions). As these schemes preserve gauge invariance, our following discussion about the reformulations in terms of gauge invariance is largely independent of the type of employed lattice fermions.

### 4.1. Lagrangian Formulation

A generic site on a (d+1)-dimensional lattice is labeled by (d+1) integers n=(n0,…,nd), where each component ranges from 1 to *N*. According to Wilson’s original construction [5], the gauge field Aμ is associated with the links of the lattice, while the Maxwell tensor Fμν is defined on the plaquettes. These quantities are conveniently expressed as [7](53)Uμ(n)=eieaAμ(n),Uμν(n)=eiea2Fμν(n),
where Uμ∈G are the link variables connecting n→n+μ^ for μ∈{0,…,d}, and Uμν(n) are the plaquette variables. These variables are graphically represented in Figure 2. The links correspond to the discretized version of Equation (49) along the line n→n+μ^. The discretized form of the Maxwell tensor can be written as(54)aFμν(n)≡Aν(n+μ^)−Aν(n)−Aμ(n+ν^)+Aμ(n).
Here, *e* and *a* are the coupling constant (electric charge) and the lattice spacing, respectively. For simplicity, we fix a=1 and restore it only in the continuum limit.

For the specific case of G=U(1), the action is given by(55)S=SG[Uμν]+Sfermions[ψ,ψ¯,Uμ],
where(56)SG=1e2∑P1−12(Uμν+Uμν†)
is the pure gauge contribution, with the summation extending over all plaquettes *P*. The term Sfermions represents the interaction with the matter fields, and its specific form depends on the chosen discretization scheme for the fermions.

### 4.2. Hamiltonian Formulation

Building on the Lagrangian formulation introduced in the previous subsection, the electric and magnetic fields now correspond to operators acting within a defined Hilbert space [43]. The link operator Uμ is the fundamental degree of freedom, related to the vector potential through Equation (53). The electric field operator, which acts on the link connecting the site n to the site n+μ^, is denoted as Eμ(n). This operator is canonically conjugate to the vector potential, and therefore it exhibits non-trivial commutation relations with the link operator Uμ(n) defined on the same site, a relation expressed as(57)[Uμ(n),Eν(n′)]=−δμ,νδn,n′Uμ(n),[Uμ†(n),Eν(n′)]=δμ,νδn,n′Uμ†(n),
while all the other commutators vanish. Using the definitions provided in Equation (53), the Hamiltonian of the system can be expressed in terms of spatial plaquette operators Uμν and electric operators Eμ. This representation is commonly referred to as the Kogut–Susskind (KS) Hamiltonian [6]. Formally, the Hamiltonian takes the form H=Hg+Hm, where(58)Hg=e22∑n,μEμ2(n)−14a2e2∑P(Uμν+Uμν†)
represents the pure gauge field contribution, and Hm denotes the matter component, whose explicit form depends on the chosen discretization of lattice fermions. In the absence of matter fields, Equation (58) describes the KS Hamiltonian for a pure U(1) Abelian LGT. The KS Hamiltonian preserves gauge invariance, meaning it commutes with the local operators:(59)G(n)=∑μ[Eμ(n)−Eμ(n−μ^)],[H,G(n)]=0.
Among the eigenstates of *H*, the physical states |ψ〉 are those satisfying the Gauss law constraint G(n)|ψ〉=0, for all the lattice sites n. It is worth noting that additional terms consistent with gauge invariance—preserving the commutation relations outlined above—can be incorporated into the Hamiltonian. Notice that in the electric basis, the sum of plaquette operators acts as a kinematic term that interchanges electric configurations. Additional potential terms that are diagonal in the electric basis can be introduced without breaking gauge invariance. A notable example is provided by the the Rokshar–Kivelson (RK) Hamiltonian [44].

The condition in Equation (59) defines the gauge-invariant Hilbert space of the system. However, it should be emphasized that the physical Hilbert space is infinite-dimensional due to the continuous nature of the U(1) gauge group. A strategy to address this point is via the introduction of quantum link models (QLMs), which replace the Wilson operators Uμ with discrete quantum variables that reside on the lattice links, referred to as *quantum links* [45,46,47].

These models provide an alternative to the KS Hamiltonian formulation of LGTs, descending from Wilson discretization: even though they have a finite-dimensional Hilbert space, they fully preserve the gauge symmetry of the original theory, at the cost of introducing non-unitary operators on the lattice links, as will be commented on later in the section. The finite number of local degrees of freedom, combined with the local symmetry, makes QLMs well suited for quantum simulation of LGTs, as quantum spins are more naturally implemented in quantum platforms. Finally, while the standard Wilson formulation of LGTs can be recovered in the large-spin limit [48,49,50,51], QLMs offer a broader set-up where novel phases may emerge, providing a richer phenomenology beyond conventional gauge theories [52,53,54,55,56]. In the following part of this section, we provide a concise overview of the bosonic and fermionic implementations of QLMs.

#### 4.2.1. Bosonic Quantum Link Models

QLMs implement the commutation relations in Equation (57) by expressing the operators in terms of quantum spin variables:(60)Uμ(n)=1S(S+1)Sμ+(n),Uμ†(n)=1S(S+1)Sμ−(n),Eμ(n)=Sμz(n).
In this framework, the local Hilbert space becomes finite-dimensional. For a spin *S*, each lattice link corresponds to a (2S+1)-dimensional Hilbert space [43]. A notable difference with the Wilson formulation is that the Wilson operators associated with different links no longer commute but instead satisfy(61)[Uμ(n),Uν†(n′)]=2Eμ(n)S(S+1)δμ,νδn,n′.
This distinction leads to interesting physical phenomena [53,54,55], while still allowing the Wilson formulation to be recovered in the large-spin limit S→∞. Indeed, as can be seen from Equation (61), the unitarity of the links is restored when taking such a limit, with the variables in Equation (60) still realizing the algebra of the underlying LGTs. The connection between the large-*S* limit and the infinite-dimensional limit is an important question, relating the two formulations of the theory, which have been investigated in the case of one-dimensional QED [51,57].

For the specific case S=1/2, each link supports only two states, corresponding to electric field values Eμ(n)=±1/2. Here, the Hamiltonian simplifies to (Sμz)2=1/4, eliminating the contribution of the electric term and leaving only magnetic interactions. The Hamiltonian in Equation (58) can be further extended by including the RK term with coupling λ, leading to(62)HRK=Hg+λ∑P(Uμν+Uμν†)2.
In the specific case of d=2, only six states satisfy the Gauss law in Equation (59). Despite its simplicity, this model exhibits rich physics [43], including a close connection to the quantum dimer model [58].

An alternative representation of the spin-1/2 QLM is achieved by mapping spins to hardcore bosons. In this perspective, the signs of the electric field (+ or −) correspond to the presence or absence of a hardcore boson on the link n→n+μ^ [56]. The gauge operators are then expressed as(63)Uμ(n)=bμ†(n),Uμ†(n)=bμ(n),Eμ(n)=nμ(n)−12.
where bμ and bμ† are bosonic creation and annihilation operators, and nμ≡bμ†bμ is the boson number operator. The plaquette term is similarly written as(64)Uμν(n)=bμ(n)bν(n+ν^)bμ†(n+ν^)bν†(n)
and describes the correlated hopping of two bosons. In this representation, the RK term involves a combination of two-particle, three-particle, and four-particle interactions. While these terms are simple to express mathematically, their implementation in ultracold atomic platforms remains challenging. In the bosonic representation, the Gauss law generators in Equation (59) take the form(65)G(n)=∑μ[nμ(n)−nμ(n−μ^)].
These operators are commuting with the Hamiltonian by construction.

#### 4.2.2. Fermionic Quantum Link Models

The particle representation offers a pathway to construct GTs by introducing fermionic links [56,59]. By substituting the bosonic creation and annihilation operators in Equation (63) with fermionic operators, a novel type of gauge theory emerges. This formulation preserves the local gauge symmetry, but the physics it describes can differ due to the different statistics of the fermionic operators, now representing the Wilson links Uμ and Uμ†. Notably, in two spatial dimensions, fermionic and bosonic QLMs are equivalent. However, in three dimensions, they define distinct models with unique physical features [56,59].

In the fermionic approach, the two-dimensional local Hilbert space is spanned by the states |0〉 and |1〉=cμ†(n)|0〉, where cμ†(n) creates a fermion on the links originating at the site n. The Wilson and electric field operators are defined as(66)Uμ(n)=cμ†(n),Uμ†(n)=cμ(n),Eμ(n)=nμ(n)−12,
where nμ(n)≡cμ†(n)cμ(n) is the fermion number operator on the link. These definitions satisfy the commutation relations in Equations (57) and (61). However, due to the fermionic statistics, the Wilson operators obey anticommutation relations.

While fermionic QLMs have been studied less extensively than their bosonic counterpart, they offer intriguing possibilities for discovering new phases in LGTs. Additionally, for LGTs in two dimensions, experimental platforms based on ultracold atoms may find the fermionic representation of plaquette interactions advantageous, as they provide an alternative method for realizing the same physics.

### 4.3. Elitzur’s Theorem

After the reminders about GTs and MF methods, we are now ready to state the crucial theorem connecting the two topics. The two examples presented in Section 2, i.e., the Ising model and the BCS theory, deal with the spontaneous symmetry breaking of a global symmetry, broken by the collective behavior of a large number of degrees of freedom of the system. For GTs, in the continuum and on the lattice, the quantum fluctuations delocalize the ground-state wave function over all possible configurations related by local transformations, therefore playing a key role in preventing the symmetry breaking. This fundamental result is known as *Elitzur’s theorem* [16], which can be explicitly stated as follows:

**Theorem 1.** 
*A local gauge symmetry cannot be spontaneously broken. The expectation value of any local observable that is not gauge-invariant must be zero.*


Here, we outline a sketch of the proof. We consider an LGT with a local symmetry group *G* and a local observable O(n) that is not gauge-invariant. We must show that its expectation value on physical states vanishes, that is, 〈O〉=0. Due to gauge invariance, any physical state |Ψ〉 satisfies the relation |Ψ〉=Ug|Ψ〉, where Ug is the unitary operator implementing a gauge transformation g(n)∈G. On the other hand, since the operator O is not gauge-invariant, it will transform as(67)UgOUg†=Og.
We can compute its expectation value in a general physical state, as(68)〈Ψ|O|Ψ〉=〈Ψ|Ug†UgOUg†Ug|Ψ〉=〈Ψ|Og|Ψ〉.
In a gauge-invariant theory, all gauge configurations are physically equivalent. Therefore, we have to average the previous statement over all possible local transformations g(n); therefore, in the end, we need to compute(69)〈Ψ|O|Ψ〉=∫Dg〈Ψ|Og|Ψ〉.
Since, by assumption, the operator is not gauge-invariant, the integration over all gauge transformations forces the expectation value to vanish, i.e., 〈Ψ|O|Ψ〉=0.

The whole argument can also be rephrased in terms of the properties of the group measure *G*, i.e., the Haar measure [7,8]. Among them, the left and right invariance under local transformations implies that we obtain a non-trivial integration only if the integrand is invariant under *G*, a statement which is equivalent to Elitzur’s theorem.

## 5. Reformulations in Terms of Gauge Invariants

The gauge symmetry is not properly physical, since it has no observable consequences to look at, as happens in the case of global symmetries. Formally, it represents a redundancy in our description of the system, in a way to describe the theory in terms of a local and causal Lagrangian. This is particularly important in the context of quantum field theories (QFTs), as non-local theories may have poles in the scattering matrix that are not associated with physical particles (with “physical”, we mean that these particles do not belong to the Hilbert space of the underlying field theory), meaning that the theory is non-unitary. Therefore, from the conventional field-theoretical point of view, working with a redundancy is advantageous, since it simplifies the computations, and at the same time, it is consistent with the properties that we want to preserve in the realm of QFT [1,2,3].

Besides the conventional writing in terms of the gauge potential there is another way to study GTs, relying on their formulation directly in terms of gauge-invariant variables. In the pure gauge case, this happens if we try to quantize the theory directly in terms of the field strength tensor Fμν, instead of the gauge potential Aμ. This procedure usually introduces two main difficulties: the Lagrangian contains non-local terms, and the dynamics of the variables are moved to the interaction terms [2]. Despite these difficulties, a crucial point is that this can be particularly helpful for the construction of consistent analytical approximation schemes, such as strong coupling expansions or MF methods [11].

In this section, we provide a summary of the main approaches presented in the literature to reformulate GTs in terms of gauge-invariant degrees of freedom, pointing out the possible advantages and drawbacks of the various methods. We distinguish three principal strategies developed over the years. The first one relies on the recombination of matter and gauge fields to rewrite the theory in terms of gauge-invariant variables; the second one regards the so-called *field strength reformulations*, entirely based on the substitution Aμ→F[Aμ] mentioned earlier, while the third one is about the *Wilson loop reformulations*, where the fundamental blocks are the Wilson lines introduced in Equation (52).

### 5.1. Recombinations of Degrees of Freedom

Historically, Dirac was the first to explore gauge-invariant reformulations of Abelian GTs [60]. He initially proposed using the field(70)Ψ(x)≡eiC(x)ψ(x),C(x)≡e∂iAi∇2
as the gauge-invariant redefinition of fermions in QED. The operator ∇−2 is the inverse of the Laplacian, whose explicit representation can be given in terms of the associated Green’s function cL(x−x′), satisfying the equation ∇2cL(x−x′)=δ(x−x′), where δ(x−x′) is the Dirac delta. This operator is intrinsically non-local, as it is the inverse of a differential operator. The object Ψ(x) represents an electron dressed with its Coulomb field: the reformulated theory has gauge-invariant operators associated with these dressed particles. Physically, we can interpret these particles as electrons surrounded by a photon cloud, given that the field Ψ(x) in Equation (70) is non-local (and even non-covariant).

A second attempt followed a few years later, when Mandelstam proposed a reformulation of QED not relying on gauge potentials [61]. His purpose was to directly use the electromagnetic field Fμν, without introducing Aμ, to show that the usual schemes to quantize QED could be derived from a gauge-independent formalism. The set of fundamental gauge-invariant variables is(71){Fμν,Ψ,Ψ*},Ψ(x)≡ψ(x)e−ie∫−∞xdξμAμ(ξ),
and we observe that Ψ(x) is again non-local and path-dependent. This last property is related to the arbitrariness in the choice of the phase factors in the field operators. Moreover, as the Fμν is now a fundamental variable, the inhomogeneous Maxwell equations must be imposed as consistency conditions, as they are not automatically satisfied. This is because, in this formulation, Fμν has no memory of its structure in terms of the gauge potential. This is a non-trivial point, and will be important in the context of the field strength reformulations.

From the early 1980s onward, a series of papers came out with the aim of establishing a precise quantization program, reformulating the physical action of QED and deriving the generating functional of the quantum theory solely in terms of gauge-invariant variables [62,63,64,65,66,67]. Specifically, scalar QED and SU(2) LGT in the presence of bosonic matter fields were investigated in [62]. The main idea was to introduce the gauge invariants of the corresponding continuum theories to rewrite the Lagrangians and derive the associated dynamics. In the continuum, this change of variables was applied to bosonic matter fields [63] and later to classical [64] and quantum [65,67] electrodynamics. In particular, matter fields were combined into new bosonic fields. In the case of 1+1 dimensions, with the Schwinger model, it was shown [66] that the construction is related to the bosonization of the original theory [68,69].

Subsequently, in [48], non-Abelian SU(N) QLMs were reformulated in terms of the so-called *rishons*, i.e., variables encoding the fermionic constituents of the non-Abelian gauge fields. Despite these variables not being explicitly gauge-invariant, they can be recombined with the color index of the fermions (or that of the neighboring links, in the pure gauge cause) to construct gauge invariants with which to reformulate the theory [70,71].

In the last decade, SU(2) LGTs with fundamental fermions were studied and reformulated via the so-called *loop–string–hadron formulation* [72]. This allows for a description of the dynamics of the theory in terms of local and physical observables, using strictly SU(2) gauge-invariant variables at the cost of introducing extra lattice links and an Abelian Gauss law. In [73,74,75,76], the problem is addressed by making use of dual formulations for the case of U(1) gauge symmetry, with particular motivation for the implementation of GTs in quantum devices.

### 5.2. Field Strength Reformulations

The purpose of this class of reformulations is to obtain, even in the case of pure GTs, a theory entirely written using Fμν. However, since not all components of this tensor are independent, the reformulated model will be constrained. In this spirit, Mandelstam was the first to consider using Fμν to quantize QED, but we inserted his proposal in the previous discussion, as he recombines gauge and matter degrees of freedom to obtain the set of variables in Equation (71).

In the late 1970s, Halpern derived the inversion Aμ→Fμν(Aμ) for QED in d=3, using a fully fixed axial gauge [77]. This change of variables can be schematically summarized in the following equations for the generating functional(72)ZA=∫DAδ[F(A)]e−14∫F2(A)⟹ZF=∫DFδ[I(F)]e−14∫F2.
Here, ZA is the starting generating functional in terms of the gauge potential, constrained with the gauge-fixing condition F(A)=0. On the right-hand side, the reformulated functional ZF is expressed using the field strength, and is now constrained with the Bianchi identities I(F)=0. The main advantages of this formulation include the absence of constraints on the state of the theory and an easier way to construct confining states in the reformulated theory.

Later in the 1980s, further attempts were made along these lines within the coordinate gauge, both in the Lagrangian and Hamiltonian formulations [78,79]. The resulting theory still exhibits a non-local action in terms of Fμν, constrained with a set of restricted Bianchi identities, similarly to Equation (72). In the Hamiltonian approach, these constraints coming from the Lagrangian formulation need to be complemented by extra conditions, to properly construct the quantization of the theory [79]. To the best of our knowledge, these papers were the first to introduce the *reconstruction theorem*, i.e., the set of conditions that Aμ and Fμν[Aμ] must satisfy in a way to be in one-to-one correspondence. We observe that a similar concept also appears in the series of papers [62,63,64,65,66,67], where it is generalized to the one-to-one correspondence between classes of generic gauge equivalent configurations and sets of gauge-invariant degrees of freedom.

### 5.3. Wilson Loop Reformulations

The last class of reformulations employs Wilson loops as fundamental variables of GTs. The first attempt in this direction, for both Abelian and non-Abelian cases, was made in the 1980s by Giles, with the aim of reconstructing the gauge potentials from a complete set of Wilson loops [80]. Here, it was proved that the Wilson loops, together with a specific class of kinematical constraints called *Mandelstam constraints*, provide sufficient conditions to reproduce the underlying local gauge theory. These constraints are necessary because the Wilson variables are non-local and form an overcomplete set with respect to the gauge potential Aμ. Moreover, it was shown that a reconstruction theorem also holds in this framework: the gauge fields can be uniquely reconstructed from the Wilson lines, up to gauge transformations. This equivalence theorem can be stated as(73)A/G≃{W(γ)}/M,
where the left-hand side represents the usual description in terms of gauge potentials and gauge transformations (A is the space of gauge orbits, G the set of gauge transformations), while the right-hand side describes the set of Wilson lines W(γ) associated with the path γ, subject to the Mandelstam constraints M [81].

The features of the space of all Wilson loops are well discussed in [82], where it is shown that this group is non-Abelian and non-locally compact, meaning that even at the local level, it is very large. The Mandelstam constraints represent non-linear algebraic equations to be solved for the Wilson lines, whose form depends on the gauge group features. In [82], their explicit form in the U(1) and SU(N) classical cases is discussed. In the lattice framework, the gauge-invariant states can be labeled by closed paths of the lattice links. The main challenge is to find an efficient way of isolating a set of independent loop states.

Several reasons have hindered the development of a consistent loop reformulation for GTs over the years [2,81,82]. In addition to difficulties arising from the quantization procedures and the complexity of the Mandelstam constraints, the complicated structure of the loop space makes it challenging to determine which part is most physically relevant. Nevertheless, the formulation in terms of independent loop variables opens new possibilities in the analysis of LGTs. First, the theory is reformulated using gauge invariants; secondly, it potentially leads to the construction of meaningful, intrinsically gauge-invariant analytical approaches, such as analytical loop perturbation theory, strong coupling and mean field expansions [11,82].

## 6. Mean Field Method Applied to Lattice Gauge Theories with Compact Gauge Group

LGTs can be considered, in every respect, statistical mechanics models on a lattice, making them amenable to analytical techniques commonly used in that field, such as series expansions and MF methods [8,11]. These approaches provide powerful tools for obtaining approximate results. However, their application to LGTs requires careful consideration of the local nature of the gauge symmetry and its implications for the choice of variables used to formulate the theory. This challenge has been explored in the literature, leading to different conclusions depending on the formulation of the underlying theory. In this section, we present an overview of the application of MF theory to pure LGTs with a generic compact gauge group and discuss different solutions to make this approximation consistent with the gauge invariance of the theory.

We start by applying the variational argument presented in Section 2 to the pure gauge action in Equation (56), in generic dimensions *d* and with compact gauge symmetry groups *G*, with a source term of the form(74)Sh[U]=h∑μ(Uμ+Uμ†)
for all the links of the lattice [11]. This results in the modified action(75)SG→S˜G−h∑μ(Uμ+Uμ†),S˜G≡1e2∑P1−12(Uμν+Uμν†)+h∑μ(Uμ+Uμ†)
In complete analogy to what is presented in Section 2.1, we first compute the partition function and free energy contribution of the source term, which plays the same role as the paramagnet Hamiltonian in the case of the Ising model. Performing all the computations at the level of a single link, we have(76)Zh=∫eh(Uμ+Uμ†)dUμ≡c(h)⇒Fh=−logc(h).

At the same time, we can compute the expectation value(77)〈Uμ+Uμ†〉h=1Zh∫(Uμ+Uμ†)eh(Uμ+Uμ†)dUμ≡t(h).
The evaluation of the integrals defining c(h) and t(h) depends on the nature of the gauge group, being the integration over the Haar measure of *G*. For example, in the case of the U(1) gauge group, where the link is essentially a phase, they are related to modified Bessel functions of the first kind. The generalization to arbitrary continuous gauge groups results in a redefinition of hyperbolic functions.

To apply the MF argument, we need to compute 〈S˜G〉h, resulting in(78)〈S˜G〉h≡1c(h)∫S˜Geh(Uμ+Uμ†)dUμ=1c(h)∫∑μh(Uμ+Uμ†)−∑P12e2(Uμν+Uμν†)eh(Uμ+Uμ†)dUμ+const.
To perform the computation, as in the case of the Ising model and the BCS theory, we assume isotropy and translational invariance, meaning that the expectation value of Uμ+Uμ† is the same independently of the direction in the space–time lattice. Neglecting the constant term due to the normalization of SG, the first term in the right-hand side of the last equation is(79)∑μ1c(h)∫h(Uμ+Uμ†)eh(Uμ+Uμ†)dUμ=dht(h),
while the second term can be written in a more explicit way, by expanding the definition of Uμν in terms of the links(80)∑P12e2c(h)∫(UμUνUμ†Uν†+H.c.)eh(Uμ+Uμ†)dUμ=d(d−1)2︸from∑P·12e2〈Uμ+Uμ†〉h4=d(d−1)2t4(h)2e2.
In the last step, we used the factorization of the expectation values of the links in the definition of the plaquette, since the computation is performed with respect to Zh, which is a single-body Hamiltonian. The final result of this computation is(81)〈S˜G〉h=dt(h)h−d−12t3(h)2e2.
Introducing β≡(2e2)−1, the equation for the free energy can be written as(82)βF=−βd(d−1)2t(h)4+dht(h)−dlogc(h),
The self-consistent equation for *h* is obtained through the differentiation(83)∂(βF)∂h=0⇒h=2(d−1)βt3(h).
We observe that, since t(h=0)=0, the trivial solution h=0 is always present, and it is the only one for strong couplings, as shown clearly in the right plot of Figure 3 for the case of G=U(1), in d=3. Moreover, it is always a local minimum for the MF free energy in Equation (82). For values of the couplings that are progressively smaller, we reach a critical value β≈βMF for which the local minimum at h=0 is accompanied by another minimum for a non-trivial value of the MF, h≠0. When β>βMF, the minimum of the free energy abruptly jumps from h=0 to h≠0. Therefore, the MF solutions of Equation (83) predict a first-order transition, and this result also holds for other gauge groups and, as usual in MF theory, is reliable only in the large-dimensional limit. As an example case, we plot the MF solutions for the U(1) gauge group, always in d=3, in the left plot of Figure 3.

A crucial observation is that, in this form, the MF solution violates Elitzur’s theorem stated in Section 4.3. This is in contrast with Equation (74): a non-trivial solution to the MF equation would imply 〈Uμ〉≠0, but we know that 〈Uμ〉=0 because the link variable is not gauge-invariant.

This problem may be faced in three different ways:Perform the computation in a given gauge. This, of course, will give a different result for each possible gauge choice. For example, in the axial gauge, where the temporal links are fixed to the identity and are no longer dynamical, this restores Elitzur’s theorem, but the price to pay is having long-range correlated spatial links along the temporal direction [83];Use a generalized MF procedure [11], where the source term is chosen as(84)Sh[U]∼∑μ,νhμν(Uμν+Uμν†),
with hμν being a square matrix for each link variable. However, the general solutions with this method are basically impossible to find. With a guided ansatz of the form hμν=hgigj−1,gi∈G, we recover the original MF picture of Equation (83), without violating Elitzur’s theorem in the intermediate steps [84];Reformulate the theory in terms of gauge-invariant variables. This allows for the choice of any possible type of order parameter, without violating Elitzur’s theorem. However, depending on the reformulation, we have different advantages and disadvantages that we are going to comment on.

### Comparisons Between Reformulations and Final Considerations

On the lattice, the first step is to change from links Uμ to plaquette variables Uμν, which was firstly accomplished by Batrouni [85]. When performing this change of variables, we obtain a Jacobian that enforces the Bianchi identity as a constraint for the new variables. Once the plaquettes are introduced, they are gauge-invariant objects by construction and can be taken as order parameters for consistent MF expansions. More recently, it was shown that the MF approximation can be further improved by determining the self-consistent mean distributions for the plaquette variables [86]. This approach works again at zero temperature, and has some disadvantages, even if it solves the inconsistency with gauge invariance. First of all, it does not straightforwardly allow the computation of non-local quantities at the MF level. Secondly, it is particularly difficult to extend to the case of non-Abelian GTs.

We also observe that the reformulation in terms of Wilson loops, despite being consistent with Elitzur’s theorem, is particularly hard to analyze using MF theory, and we are not aware of attempts in this direction. This is due to the presence of Mandelstam constraints, discussed in Section 5.3, which are more complicated to handle with respect to the Bianchi identity.

Finally, we mention that LGTs can also be reformulated in terms of Polyakov loops, to obtain a description in terms of effective line actions. After the reformulation, consistent MF procedures can be developed for the effective theory at any temperature [87].

## 7. Conclusions

Mean field (MF) theory is a way to obtaining approximate information about the behavior of interacting systems, and is widely used as an initial tool for the analysis of lattice models. While MF theory is known, in some cases, to produce quantitatively and qualitatively inaccurate results, it provides, at the same time, a basis for more sophisticated approximations. When applied to lattice gauge theories (LGTs), a key aspect is not only obtaining qualitatively reliable results but also enforcing the gauge invariance of the theory in a way that the computed quantities are not dependent on the gauge choice. The effectiveness of the MF depends on the specific reformulation of the theory, i.e., the choice of variables used to express it. Therefore, when an LGT is treated with an approximate method such as MF, it is crucial to clarify what is the most convenient reformulation that retains gauge invariance while achieving the best possible result. Another related question is which reformulation, when treated with MF, provides the best starting point for further refinements.

Motivated by these questions and the relative lack of recent literature on the MF treatment of reformulated LGTs, we have presented a discussion of MF approaches to LGTs in this paper. Another motivation arises from the fact that MF studies of LGTs are scattered across a long period, starting from the 1950s. We believe it is useful to discuss the challenges of MF approaches in LGTs, connecting both older and more recent results.

After an overview of the MF theory in statistical mechanics and many-body systems, we studied its application to pure LGTs with a generic compact gauge group. We then reviewed the existing literature on the topic, discussing different reformulations of LGTs and how they influence the MF approximation. We presented a comparative discussion of how to perform the MF approximation in the different reformulations. Advantages and disadvantages has been discussed, with mention of possible perspectives, also in view of the ongoing progress in the field of quantum simulations of LGTs with ultracold atoms. 

## Figures and Tables

**Figure 1 entropy-27-00250-f001:**
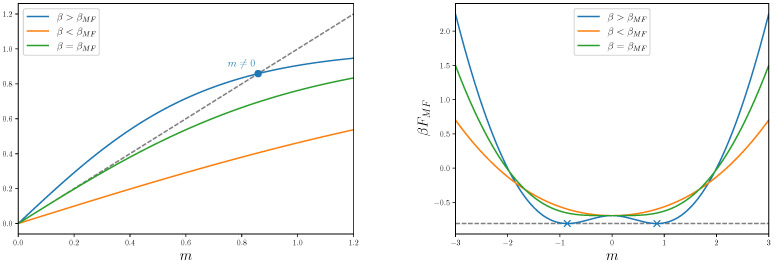
(**Left plot**) Solutions to the Equation (15), in terms of the magnetization *m* and three different values of β, β=−βMF/2 (orange line), β=3βMF/2 (blue line) and β=βMF (green line). The grey dashed line represents the function y=m, while the blue dot highlights the non-trivial value of the magnetization, the solution of Equation (15), for β>βMF. (**Right plot**) Mean field free energies in Equation (16) as a function of the magnetization *m*. The values of β and the color code are the same as the (**Left plot**). The horizontal dashed grey line represents the value of βFMF in correspondence with the non-trivial solutions, m≠0, highlighted with blue cross markers.

**Figure 2 entropy-27-00250-f002:**
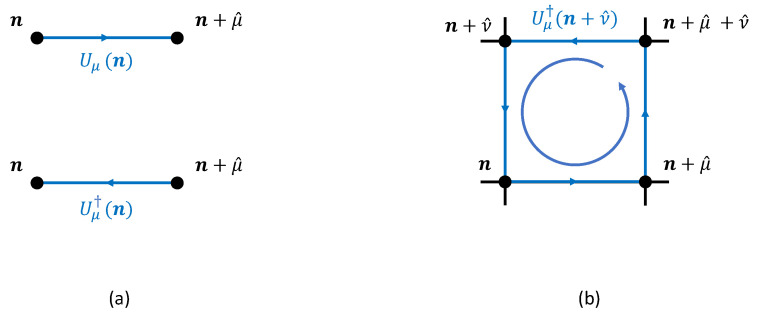
(**a**) Graphical representations of link variables Uμ(n), Uμ†(n). (**b**) The contribution Uμν(n) of an elementary plaquette with base site n, in the μ−ν plane.

**Figure 3 entropy-27-00250-f003:**
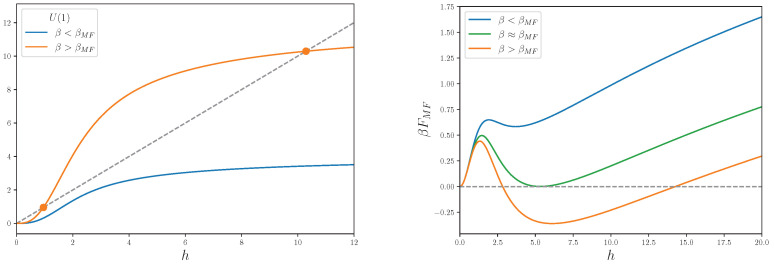
(**Left plot**) Solutions to Equation (83), as functions of the mean field *h* and two different values of β, β<βMF (blue line) and β>βMF (orange line). The grey dashed line represents the function y=h, while the orange dots depict the non-trivial values of the mean fields, solutions of Equation (83), for β>βMF. (**Right plot**) Mean field free energies in Equation (82) as a function of the mean field *h*, for three values of β, including β≈βMF. The horizontal dashed line represents the minimum at h=0, in correspondence with which βFMF=0.

## Data Availability

No new data were created or analyzed in this study.

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
