# Peer review of "Mean Field Approaches to Lattice Gauge Theories: A Review"

_entropy, 2025, doi:10.3390/e27030250_

Round 1
Reviewer 1 Report
Comments and Suggestions for Authors
This review paper discusses the potential use of mean field methods in addressing gauge theories, including their lattice formulation. It begins with an introduction to mean field theory, followed by applications to spin models and many-body systems. In the latter case, the authors explain how mean field theory can capture key features of superconductivity.
The paper then introduces gauge symmetry in quantum field theory, with a focus on the concept of parallel transport. This framework is used to present lattice gauge theories in both Lagrangian and Hamiltonian formulations. Elitzur’s theorem is discussed, emphasizing the crucial point that gauge non-invariant quantities have zero expectation values. Consequently, to apply mean field theory to gauge theories, a reformulation in terms of gauge-invariant quantities is necessary. The paper explores several approaches to formulating lattice gauge theories using gauge-invariant variables.The final section is dedicated to potential applications of these methods to pure lattice gauge theories.
I found this paper valuable for summarizing an interesting subject and compiling relevant scientific literature. In my opinion, it is well-written and suitable for inclusion in the special issue ``Foundational Aspects of Gauge Field Theory''. Therefore, I recommend it for publication.
Author Response
We thank the Referee for having accepted to review our work and the time spent on it, and for
the positive evaluation of our paper.

Reviewer 2 Report
Comments and Suggestions for Authors
The present work represents a review on mean-field theory, exemplified by the Ising model and BCS theory, with subsequent applications to gauge theory with compact gauge groups formulated in the Euclidean continuum and on Euclidean lattices.
The review dwells (in a quite sloppy way) on well known results. The presentation needs to be improved according to the remarks I added in the attached pdf of the ms. The formulations of gauge theories in terms of gauge invariant variables and in gauge fixed versions are well known. Their essential aspects (consistency conditions: Bianchi, inhomogeneous field equations, Gauss' law) are mentioned in passing only and should be made much more explicit.
New to me are bosonic and fermionic Quantum Links Models which relate link variables to quantum spin variables (irreducible SU(2) reps). Here, it needs to be pointed out what these models conceptually strive to achieve and why the limit S->oo yields back the ordinary link variable of the gauge theory.
Sec. 6 is rather foggy. I would welcome if (69) were derived in the review.
Finally, essential literature is missing, see pdf.
On the whole this review can be published provided my above points and all those in the pdf are addressed satisfactorily and the authors put more effort into really teaching the key concepts, say of Quantum Link Models and the Mean-Field Approach to gauge theory models.

Author Response
We thank the Referee for the comments and criticisms, and for raising very useful points. We have taken the feedback seriously and accordingly modified the manuscript, doing our best to improve the clarity and presentation of the review.
- Referee: The review dwells (in a quite sloppy way) on well-known results. The presentation needs to be improved according to the remarks I added in the attached pdf of the ms. The formulations of gauge theories in terms of gauge invariant variables and in gauge fixed versions are well known. Their essential aspects (consistency conditions: Bianchi, inhomogeneous field equations, Gauss' law) are mentioned in passing only and should be made much more explicit.
Authors: We agree that we have been a bit synthetic and passed over important aspects of gauge theories, like the Bianchi identities for the field strength tensor, the inhomogeneous field equations of the vector potential and the resulting Gauss law. For this reason, we added a Subsection in the resubmitted version of the manuscript, called “Equations of motion and Bianchi identity” (Subsec. III A), to address the main point of the Referee. We hope that, with this addition, our presentation is improved, and the manuscript is more readable.
- Referee: New to me are bosonic and fermionic Quantum Links Models which relate link variables to quantum spin variables (irreducible SU(2) reps). Here, it needs to be pointed out what these models conceptually strive to achieve and why the limit S->oo yields back the ordinary link variable of the gauge theory.
Authors: We argument about these points in Subsection IV B, when explaining and introducing the Hamiltonian formulation of lattice gauge theories. We added a paragraph about the relationship between the Wilson discretization and the quantum link models, clarifying why these are important both theoretically and experimentally. Moreover, the large spin limit is of key importance, and we thank again the Referee for pointing out that. In the version we submitted, the relation between the link variables and the quantum spins does not allow for such extrapolation, as normalization factors were neglected. We fixed this (see Eqs. (60) and (61) of the resubmitted version) and commented immediately below them why we recover the full lattice gauge theory when enlarging progressively the local Hilbert space of the quantum link formulation. The relation between the physical properties of the two theories for large spin, however, is not straightforward, and has been the object of different studies in the literature that we cite in our discussion.
- Referee: 6 is rather foggy. I would welcome if (69) were derived in the review.
Authors: We agree with the Referee that, in the submitted version, the mean field free energy comes out without a deep explanation. For this reason, following the presentation of Section II and the related examples, in the revised version we exemplify all the steps leading to Eq. (69), commenting on all the terms coming out from the application of the Jensen inequality. We also rephrased the discussion about the order of the transition predicted from the theory, complementing it with a new picture (right plot of Fig. 3). We hope that with these corrections the Section has gained clarity, shading light on some aspects that we did not discuss in detail in the first stage of the submission.
In the attached PDF we include as well a list of changes made in the resubmission.

Round 2
Reviewer 2 Report
Comments and Suggestions for Authors
The authors have revised the ms according to my suggestions. Therefore, the ms can be published in Entropy as it stands.